# Changes in the Choroidal Thickness following Intravitreal Bevacizumab Injection in Chronic Central Serous Chorioretinopathy

**DOI:** 10.3390/jcm11123375

**Published:** 2022-06-13

**Authors:** Yoo-Ri Chung, Su Jeong Lee, Ji Hun Song

**Affiliations:** Department of Ophthalmology, Ajou University School of Medicine, Suwon 16499, Korea; cyr216@hanmail.net (Y.-R.C.); 111829@aumc.ac.kr (S.J.L.)

**Keywords:** bevacizumab, central serous chorioretinopathy, choroidal thickness

## Abstract

We evaluated the effect of intravitreal bevacizumab injection (IVB) on choroidal thickness, and studied its association with the therapeutic response in chronic central serous chorioretinopathy (CSC). The clinical features of 78 eyes with chronic CSC treated with IVB from October 2014 to June 2020 were retrospectively evaluated. Visual acuity (VA), central retinal thickness (CRT), and sub-foveal choroidal thickness (SFCT) were analyzed at baseline, 1 month following initial IVB, and the last follow-up examination. Cases showing complete recovery (resolved eyes; *n* = 60) were compared with those with persistent subretinal fluid (refractory eyes; *n* = 18). The relationship between the potential risk factors and subretinal fluid resolution was examined using logistic regression. SFCT was significantly decreased along with the CRT following IVB at the resolved state. SFCT reduction following 1 month of IVB was notably greater in the resolved eyes. The association of refractory eyes with hypertension (*p* = 0.003) and a thinner baseline SFCT (*p* = 0.024) was significant. In most of the patients with chronic CSC, VA and CRT remarkably improved following treatment with IVB. Early changes in the SFCT following IVB were associated with the therapeutic response. Patients with hypertension and a thinner baseline SFCT could be unresponsive to IVB.

## 1. Introduction

Central serous chorioretinopathy (CSC) is a self-limiting disease characterized by idiopathic serous detachment of the neurosensory retina [1]. Compared with the healthy population, patients with CSC show a significant sympathetic–parasympathetic imbalance, probably related to choroidal blood flow modulation [1]. Moreover, individuals at a higher risk for developing CSC are potentially those under corticosteroid use, as well as those with a “type-A” personality, which is associated with greater emotional stress. Although the pathogenesis of CSC is complex and not completely understood, choroidal hyperpermeability and retinal pigment epithelium (RPE) dysfunction have been suggested as possible pathologic mechanisms [1].

With the development of enhanced depth imaging (EDI) optical coherence tomography (OCT), multiple studies have reported greater choroidal thickness in CSC eyes compared with that in healthy eyes [2,3]. The choroid of both the involved and uninvolved fellow eyes were found to be thicker in patients with CSC than in healthy controls [3,4,5]. Moreover, studies also reported greater sub-foveal choroidal thickness in CSC eyes with a higher vascular luminal ratio [6]. Significant reduction in the stromal area of the choroid has been noted in patients with CSC, along with increased luminal area (i.e., higher vascular luminal-to-total choroidal ratio). Choroidal vascular area (luminal area) positively correlated with choroidal thickness in CSC, suggesting that chronic venous congestion and subsequent stromal tissue atrophy might be involved in the pathophysiology of CSC [6].

Observation is initially recommended for acute CSC, which is usually a self-limiting disease [1]. However, there is no consensus on the precise timing of intervention for chronic, atypical, or recurrent CSC, which are more complex subtypes of CSC that can result in permanent damage to the photoreceptors and irreversible visual impairment owing to persistent subretinal fluid [7,8]. Half-dose photodynamic therapy (PDT) and intravitreal injection of anti-vascular endothelial growth factor (VEGF) agents have been used to manage these vision-threatening subtypes of CSC [9,10,11,12]. PDT can potentially reduce the underlying choroidal hyperpermeability and congestion; however, a risk of complications, such as permanent RPE atrophy or choriocapillaris hypoperfusion, exists [10,13]. Among multiple anti-VEGF agents, intravitreal bevacizumab injection (IVB) has been widely used without significant complications in an off-label manner [14,15].

The beneficial effect of IVB was previously reported on the visual and anatomical improvement of patients with chronic, atypical, or recurrent CSC over a follow-up period of 1 year [12]. IVB can reduce choroidal hyperpermeability followed by a subsequent reduction in choroidal thickness [16]. Considering CSC, previous studies reported an increased choroidal thickness in active CSC and a reduced choroidal thickness along with resolution of subretinal fluid following various treatment modalities [17]. However, the changes in choroidal thickness following IVB and its clinical implication in the treatment of CSC have not been fully investigated. Our study aimed to evaluate the effects of IVB on choroidal thickness and its association with the therapeutic response in patients with chronic CSC.

## 2. Materials and Methods

In this retrospective interventional case series, the medical records of patients diagnosed with chronic CSC who underwent IVB between October 2014 and June 2020 in Ajou University Hospital, Suwon, Korea were reviewed. This study complied with the Declaration of Helsinki and was approved by the Institutional Review Board of Ajou University Hospital, Suwon, Korea (IRB No.: AJIRB-MED-MDB-19-313). The need for informed consent was waived owing to the retrospective nature of the study by the Institutional Review Board of Ajou University Hospital. Patient demographics were obtained from the medical records, including age, sex, history of hypertension, and smoking status.

The diagnosis of CSC was formulated based on multimodal imaging results as described previously [12]. Briefly, chronic CSC was defined as CSC with subretinal fluid persisting for longer than 4 months along with generally widespread RPE changes [1,12]. Eyes with typical acute CSC and those with suspicious choroidal neovascularization or polypoidal choroidal vasculopathy on fluorescein angiography or indocyanine green angiography were excluded. Eyes with CSC that were previously treated with intravitreal injections, PDT, and/or focal laser treatment as well as eyes with other retinal pathologies or history of vitrectomy were also excluded.

IVB was performed at intervals of 4–6 weeks until no subretinal fluid was observed on OCT. Visual acuity (VA), central retinal thickness (CRT), and sub-foveal choroidal thickness (SFCT) were analyzed at the baseline, 1 month following initial IVB, at the time of complete resolution of subretinal fluid in resolved eyes, and at the last follow-up examination. CRT data were obtained from an automatically generated macular grid map of Spectralis OCT (Heidelberg Engineering, Heidelberg, Germany). The SFCT was measured manually using EDI mode, and was defined as the distance from the outer border of the hyper-reflective line corresponding to the RPE perpendicular to the chorio-scleral interface. Two independent examiners (Y.R.C. and S.J.L.) measured the SFCT at each visit, and their mean data were used for statistical analysis. Each eye was classified into two groups based on the response to the IVB treatment at the follow-up visit: eyes with resolved subretinal fluid (resolved CSC group) and eyes with refractory CSC (refractory CSC group). The resolved CSC group was defined as eyes in which the subretinal fluid was completely resolved following initial consecutive IVB, and the data until the last follow-up visit maintaining fluid-free state were included. The refractory CSC group included eyes with persistent subretinal fluid during follow-up. Datasets that support the study findings are available in Appendix A.

Statistical analyses were conducted using SPSS 23.0 (IBM Corp., Armonk, NY, USA). The interobserver reliability of the SFCT measurements was expressed as the intraclass correlation coefficient (ICC). The chi-squared test and independent *t*-test were used to compare the categorical and continuous variables between the resolved and refractory groups, respectively. A paired *t*-test was used to detect the changes in the numerical values at each time point of the study relative to baseline values within groups. A repeated measures analysis of variance (ANOVA) test was performed to verify the differences in the changes in the VA, CRT, and SFCT between the resolved and refractory eyes. Logistic regression analysis was performed to identify the factors associated with the resolution of subretinal fluid, presented as the odds ratio (OR), with a 95% confidence interval (CI) and *p*-value. A *p*-value < 0.05 was considered statistically significant.

## 3. Results

A total of 78 eyes from 76 patients (59 men and 17 women, age range 30–71 years) with chronic CSC were included in this study. Two patients had bilateral CSC. All the included patients were treatment-naïve and without macular neovascularization. Among the included eyes, 60 (77%) eyes showed complete resolution of the subretinal fluid at the follow-up visit (resolved CSC group), while 18 (23%) eyes showed persistent subretinal fluid (refractory CSC group; Table 1). There was no difference in the baseline ocular parameters between both the groups, with the exception of baseline SFCT, which was thicker in the resolved CSC group (*p* = 0.024). The mean age of the patients was 50.4 ± 8.8 years (range: 30–71 years). The prevalence of hypertension among the patients was 18.4% (14 of 76 patients). The inter-rater reliability of the SFCT measurements was excellent (ICC = 0.928, *p* < 0.001). The mean follow-up period was 13.1 ± 14.0 months. No severe adverse events associated with the IVB procedure, such as endophthalmitis or retinal detachment, were noted.

Changes in the ocular parameters 1 month following the initial IVB are summarized in Table 2. The changes in the CRT and SFCT were significantly different between the resolved and refractory CSC eyes (*p* = 0.036 for CRT and *p* = 0.047 for SFCT (repeated measures ANOVA)), while the change in the VA was not statistically significant. Among the resolved CSC eyes, the mean logarithm of the Minimum Angle of Resolution (logMAR) VA improved at 1 month following IVB (0.30 ± 0.25 (Snellen equivalent of 20/40) to 0.24 ± 0.23 (Snellen equivalent of 20/35), *p* = 0.003 (paired *t*-test)) along with significant reductions in the CRT (392.2 ± 120.3 μm to 259.1 ± 94.5 μm, *p* < 0.001 (paired *t*-test)) and SFCT (404.4 ± 102.7 μm to 383.0 ± 105.5 μm, *p* = 0.015 (paired *t*-test)). The mean period required for subretinal fluid resolution was 5.3 ± 4.6 months. However, there were no significant changes in the VA, CRT, or SFCT in the refractory CSC eyes at 1 month following IVB.

The improvement of VA remained significant in the resolved CSC eyes at the final follow-up visit (0.21 ± 0.24 (Snellen equivalent of 20/32), *p* = 0.002 (paired *t*-test)), along with a significant reduction of CRT (235.9 ± 63.5 μm, *p* < 0.001 (paired *t*-test)) and SFCT (338.7 ± 109.9 μm, *p* < 0.001 (paired *t*-test)). Representative cases are shown in Figure 1.

There were no significant changes in the VA (0.34 ± 0.44 (Snellen equivalent of 20/44), *p* = 0.366 (paired *t*-test)), CRT (341.4 ± 81.7 *p* = 0.526 (paired *t*-test)), or SFCT (349.2 ± 141.6, *p* = 0.492 (paired *t*-test)) in the refractory CSC eyes at the final follow-up visit. The mean number of IVBs required for the subretinal fluid resolution was 2.4 ± 1.6 (range: 1–8) within 5.4 ± 4.7 months (range: 1.1–26.7) in the resolved eyes, while IVB was performed 3.6 ± 3.2 (range: 1–15) times within 7.6 ± 8.0 months in the refractory eyes. During the follow-up period, significant differences were observed in the CRT and SFCT measured between the resolved eyes and the refractory CSC eyes (*p* = 0.001 for CRT and *p* = 0.027 for SFCT (repeated measures ANOVA), Figure 2).

Logistic regression analysis was performed to identify the factors associated with refractory subretinal fluid following IVB (Table 3). The results showed that older age (OR = 1.074; 95% CI, 1.007–1.146; *p* = 0.029) and the presence of hypertension (OR = 7.200; 95% CI, 2.052–25.263; *p* = 0.002) were associated with a significantly higher risk of refractory subretinal fluid. In addition, a thicker baseline SFCT was associated with a significantly lower risk of refractory subretinal fluid following IVB (OR = 0.994; 95% CI, 0.989–0.999; *p* = 0.029). In the multiple logistic regression analysis, hypertension and baseline SFCT remained significantly associated with a higher risk of refractory subretinal fluid (OR = 10.448; 95% CI, 2.238–48.771; *p* = 0.003 for hypertension and OR = 0.993; 95% CI, 0.986–0.999; *p* = 0.024 for baseline SFCT).

## 4. Discussion

In eyes with CSC, the engorgement of the choroidal vessels results in increased hydrostatic pressure of the choroid, thereby inducing subretinal fluid [1]. Although multiple interventions have been suggested for treating CSC, there is no consensus that any single treatment is more advantageous over observation or other interventions [18]. It is especially difficult to determine the efficacy of interventions in acute CSC due to its benign nature and spontaneous improvement. Meanwhile, chronic CSC is more complex due to its sequalae of visual impairment. Half-dose PDT and the use of mineralocorticoid antagonists have been suggested for CSC treatment [18]; however, permanent RPE damage can occur with PDT [10,13] and the use of mineralocorticoid antagonists has been proven ineffective for chronic CSC [19].

IVB has been considered effective in chronic CSC treatment [12,20]. Considering the association of choroidal hyperpermeability and choroidal thickness, IVB may reduce choroidal hyperpermeability followed by a subsequent reduction in the choroidal thickness [16]. Abnormal choroidal flow as seen on OCT angiography is associated with CSC disease activity [21]. Our study demonstrated a significant reduction of SFCT and CRT following IVB, especially in the eyes with resolved CSC. The reduction of SFCT was observed from 1 month following IVB to the last follow-up visit in resolved CSC eyes, while the SFCT remained unchanged in refractory CSC eyes. This positive effect of IVB on SFCT was observed in 77% of the studied cases and may suggest a decrease in choroidal congestion and disease activity owing to the anti-VEGF effect.

Previous studies have investigated the changes in SFCT following IVB in various retinal vascular diseases, such as diabetic retinopathy and neovascular age-related macular degeneration (AMD) [22,23]. The decreasing effect of IVB on choroidal thickness was noted at 1 week following IVB, which remained until 1 month following IVB in diabetic macular edema and AMD [22]. Campos et al. [23] also reported that the SFCT decreased following the administration of anti-VEGF injections, although baseline SFCT was not a significant predictor of either the functional or the anatomic response in eyes with diabetic macular edema. In PCV, anti-VEGF therapy appears to reduce VEGF-mediated exudation and choroidal vascular flow and volume, leading to reduced choroidal thickness [24,25]. Considering CSC, several studies investigating the changes in SFCT, reported increased choroidal thickness in active CSC, and reduced SFCT along with resolution of subretinal fluid [17]. Kim et al. [17] also reported that recurrence was more frequent in eyes with CSC with a small reduction in SFCT. Our data demonstrated that a thicker baseline SFCT was associated with a lower risk of refractory subretinal fluid following IVB in eyes with chronic CSC. Moreover, a decreasing effect of bevacizumab on SFCT was observed 1 month following initial IVB treatment in the eyes with resolved CSC, while no effect was found in the eyes with refractory CSC. This finding could suggest that an early reduction in SFCT following initial IVB may be an important indication towards the response to IVB in eyes with chronic CSC.

In our study, more patients with refractory subretinal fluid had hypertension. This chronic condition was also associated with refractory subretinal fluid in eyes with chronic CSC, along with a thinner choroidal thickness [1]. Hypertension has been suggested as a risk factor in developing CSC, especially in Asian populations [26,27]. Persistently high blood pressure may affect arterioles, including choroidal vessels, by inducing repeated vasoconstriction, followed by the occlusion of choriocapillaris and disruption of the blood–retinal barrier [26]. This hypothesis was suggested for the explanation of the association of hypertension and CSC occurrence; however, this also can be applied to the responsiveness of CSC to IVB as shown in this study. We were unable to investigate data pertaining to the use of antihypertensive medications in any patients with hypertension who could have been included in this study, due to its retrospective nature. This limitation implies the possible influence of CYP34A inhibition by calcium channel blockers (CCBs) [28,29]. Cortisol is metabolized by CYP3A4 [30] such that the use of antihypertensive medications, such as CCBs, might affect the association of hypertension with refractory subretinal fluid. Although CCBs are much less commonly prescribed compared with other antihypertensive treatments [31], further studies on CSC that include antihypertensive medications are needed.

The primary strength of this study was its large sample size compared with previous studies investigating the effect of IVB in chronic CSC, as well as among the studies evaluating changes in choroidal thickness following IVB in all kinds of CSC. Moreover, to the best of our knowledge, this is the first study to have investigated the changes in choroidal thickness as a predictive factor for the treatment response following IVB. This study also analyzed the associated baseline risk factors for poor responsiveness to IVB in chronic CSC. However, there are several limitations to this study related to its retrospective design. For instance, the possibility of inclusion of cases with spontaneously improved CSC exists; however, patients with acute CSC and those who improved spontaneously during the initial 3-month period following diagnosis were excluded. Thus, the change in the ocular parameters observed in this study could be considered as solely attributed to the effect of IVB. It should be noted that there are still no prospective randomized controlled clinical trials with large sample size that investigated the effect of anti-VEGF agents in CSC. Accordingly, caution should be taken in the interpretation of the results although we found a positive effect of IVB in chronic CSC. Meanwhile, a more interesting point of this retrospective study is that the change of choroidal thickness following IVB was associated with the therapeutic response in chronic CSC, and this is a potential advantage. All the patients included in this study were treated exclusively with IVB and not with PDT. The latter was not available during most of the study period because neither maintenance support for the PDT machine by its manufacturer nor import of verteporfin dye were available. However, PDT was our last option for the treatment of these patients with chronic CSC, as it has potential complications such as permanent RPE atrophy and choriocapillaris hypoperfusion. Eyes with chronic CSC are already prone to RPE atrophy. Finally, lack of choroidal vascularity index data is another limitation of this study, as this could provide additional information associated with SFCT [32]. Further prospective studies with large sample sizes are warranted to examine the long-term effects of IVB on SFCT in chronic CSC after accounting for several confounders.

## 5. Conclusions

In conclusion, SFCT was reduced along with an improvement in CRT and best corrected VA following IVB in a large number of patients with chronic CSC. Early changes in SFCT following IVB were observed in cases showing a complete recovery, and these changes were found to be associated with the therapeutic response in chronic CSC. Additionally, hypertension and a thinner choroidal thickness may be important factors associated with the poor responsiveness to IVB. Thus, hypertension and baseline choroidal thickness should be considered before deciding IVB treatment for chronic CSC management. Finally, a limited effect of IVB should be expected in patients with hypertension or in eyes with CSC showing no significant reduction in SFCT following initial IVB treatment.

## Figures and Tables

**Figure 1 jcm-11-03375-f001:**
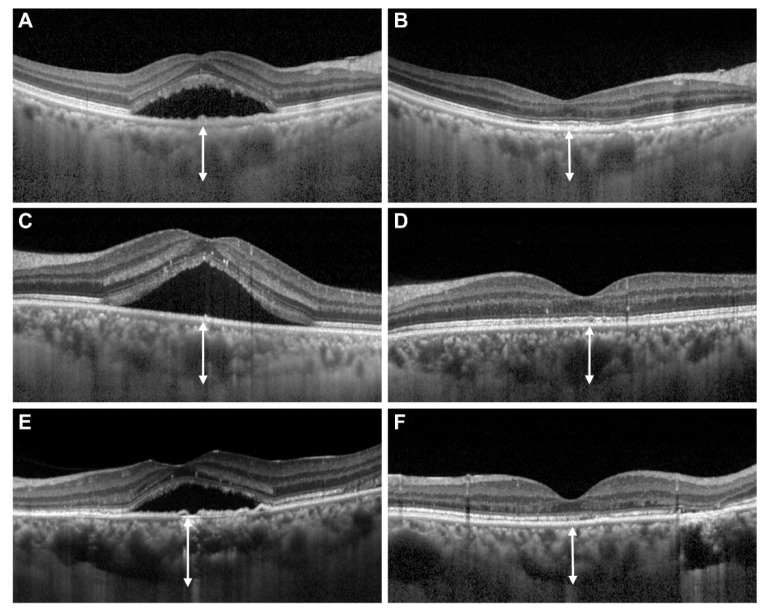
Representative cases of resolved central serous chorioretinopathy (CSC) following intravitreal bevacizumab injection (IVB). (**A**) Subretinal fluid was noted in a 63-year-old man with chronic CSC in the right eye, showing a central retinal thickness (CRT) of 485 μm and sub-foveal choroidal thickness (SFCT) of 409 μm. (**B**) Following two IVBs, the CRT and the SFCT decreased to 188 μm and 303 μm, respectively. (**C**) Subretinal fluid was noted in a 37-year-old man with chronic CSC in the left eye, showing a CRT of 537 μm and SFCT of 400 μm. (**D**) Following one IVB, the CRT and the SFCT decreased to 193 μm and 371 μm, respectively. (**E**) Subretinal fluid was noted in a 43-year-old woman with chronic CSC in the left eye, showing a CRT of 329 μm and SFCT of 401 μm. (**F**) Following monthly IVBs performed 6 times, the CRT and the SFCT decreased to 163 μm and 387 μm, respectively.

**Figure 2 jcm-11-03375-f002:**
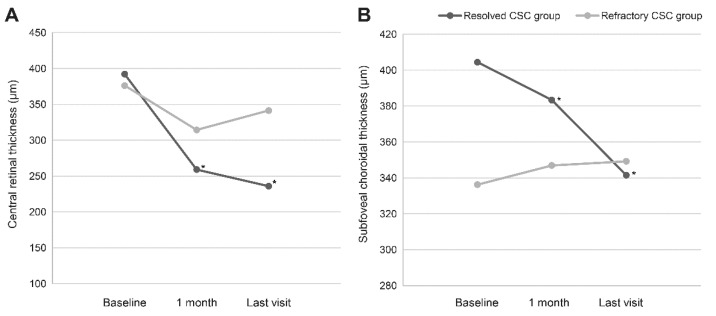
Changes in the central retinal thickness (CRT) and sub-foveal choroidal thickness (SFCT) in eyes with resolved central serous chorioretinopathy (CSC) and those with refractory CSC. We compared (**A**) the changes in the CRT and (**B**) the changes in the SFCT during the follow-up period between resolved and refractory CSC eyes and found a significant difference in the CRT and SFCT changes between the two groups (*p* = 0.001 for CRT and *p* = 0.027 for SFCT by repeated measures analysis of variance). (**A**) In the eyes with resolved CSC, a significant reduction in CRT was noted 1 month following initial IVB (*p* < 0.001 by paired *t*-test). (**B**) Similarly, a significant reduction in SFCT from baseline was noted 1 month after initial IVB (*p* = 0.015 by paired *t*-test), and at the last visit (*p* < 0.001 by paired *t*-test). * Timepoints that variables (CRT and SCFT) showed significant change from baseline (*p* < 0.05 by paired *t*-test).

**Table 1 jcm-11-03375-t001:** Baseline and ocular characteristics of included patients based on the response to the intravitreal bevacizumab injection during the follow-up period.

	Resolved CSC	Refractory CSC	*p*-Value
No. of patients	59	17	
Age (years)	50.2 ± 8.7	51.4 ± 9.1	0.627 *
Sex, male	46/59 (78%)	13/17 (76%)	0.896 ^†^
Hypertension	10/59 (17%)	4/17 (24%)	0.537 ^†^
Current smoking	16/41 (39%)	6/13 (46%)	0.648 ^†^
Follow-up period (months)	12.0 ± 12.4	16.9 ± 18.3	0.308 *
No. of eyes	60	18	
Baseline VA (logMAR)	0.30 ± 0.25	0.39 ± 0.36	0.252 *
Baseline VA (Snellen)			0.377 ^†^
20/40 or better	37/60 (62%)	9/18 (50%)	
20/50 or worse	23/60 (38%)	9/18 (50%)	
Baseline CRT (μm)	392.2 ± 120.3	376.3 ± 139.6	0.637 *
Baseline SFCT (μm)	404.4 ± 102.7	336.2 ± 133.2	0.024 *
FA leakage	31/60 (52%)	12/18 (67%)	0.291 ^†^

* *p*-value by independent *t*-test. ^†^
*p*-value by chi-square test. CRT = central retinal thickness, CSC = central serous chorioretinopathy, FA = fluorescein angiography, SFCT = sub-foveal choroidal thickness, VA = visual acuity, logMAR = Logarithm of the Minimum Angle of Resolution.

**Table 2 jcm-11-03375-t002:** Changes in the ocular parameters 1 month following initial intravitreal bevacizumab injection (IVB) based on the final response to IVB.

	Resolved CSC (*n* = 60)	Refractory CSC (*n* = 18)	*p*-Value of Changes between Groups
Baseline	59	17	
VA (logMAR [SE])	0.30 ± 0.25 [20/40]	0.39 ± 0.36 [20/50]	0.252 *
CRT (μm)	392.2 ± 120.3	376.3 ± 139.6	0.637 *
SFCT (μm)	404.4 ± 102.7	336.2 ± 133.2	0.024 *
Post-1 month state	60	18	
VA (logMAR [SE])	0.24 ± 0.23 [20/38]	0.32 ± 0.35 [20/42]	0.864 ^†^
CRT (μm)	259.1 ± 94.5	314.4 ± 97.8	0.036 ^†^
SFCT (μm)	383.0 ± 105.5	346.9 ± 159.5	0.047 ^†^
*p*-value of changes within groups			
VA (logMAR)	0.003 ^‡^	0.079 ^‡^	
CRT (μm)	<0.001 ^‡^	0.132 ^‡^	
SFCT (μm)	0.015 ^‡^	0.331 ^‡^	

* *p*-value by independent *t*-test. ^†^
*p*-value by repeated-measures ANOVA. ^‡^
*p*-value by paired *t*-test CRT = central retinal thickness, CSC = central serous chorioretinopathy, IVB = intravitreal bevacizumab injection, SE = Snellen equivalent, SFCT = sub-foveal choroidal thickness, VA = visual acuity, logMAR = Logarithm of the Minimum Angle of Resolution.

**Table 3 jcm-11-03375-t003:** Logistic regression analysis to investigate the association of potential risk parameters with refractory subretinal fluid in eyes with chronic central serous chorioretinopathy (CSC) treated with intravitreal bevacizumab injection (IVB) (*n* = 78).

Variables	OR (95% CI)	*p*-Value
Age	1.074 (1.007–1.146)	0.029 *
Sex (male)	1.065 (0.302–3.763)	0.922
Hypertension	7.200 (2.052–25.263)	0.002 *
Baseline VA (logMAR)	3.000 (0.458–19.662)	0.252
Baseline CRT	0.999 (0.995–1.003)	0.633
Baseline SFCT	0.994 (0.989–0.999)	0.029 *
Leakage at FA	1.871 (0.621–5.638)	0.266

* *p*-value < 0.05 by logistic regression analysis. CI = confidence interval, CRT = central retinal thickness, CSC = central serous chorioretinopathy, FA = fluorescein angiography, IVB = intravitreal bevacizumab injection, OR = odds ratio, SFCT = sub-foveal choroidal thickness, VA = visual acuity, logMAR = Logarithm of the Minimum Angle of Resolution.

## Data Availability

Anonymized raw data of all the study eyes included in the analysis of this study are available in Appendix A.

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
