# Peer review of "Changes in the Choroidal Thickness following Intravitreal Bevacizumab Injection in Chronic Central Serous Chorioretinopathy"

_jcm, 2022, doi:10.3390/jcm11123375_

Round 1
Reviewer 1 Report
Reject: The article has serious flaws, does not make an original contribution. Although the topic raised by the authors is interesting and the examination of the thickness of the choroid after the administration of VEGF drugs is important, the article has significant drawbacks. All the mataanalyses show that there is not enough strong evidence that injecting a-VEGF drugs does not have tremendously strong evidence of their effectiveness. Meta-analyzes showed only the efficacy of treatment with mineralocorticoid antagonists and half the dose of PDT.
Salehi M, Wenick AS, Law HA, Evans JR, Gehlbach P. Interventions for central serous chorioretinopathy: a network meta‐analysis. Cochrane Database of Systematic Reviews 2015, Issue 12. Art. No.: CD011841. DOI: 10.1002/14651858.CD011841.pub2. Accessed 15 May 2022."Low quality evidence from two trials suggested little difference in the effect of anti-VEGF (ranibizumab or bevacizumab) or observation on change in visual acuity at six months in acute CSC (mean difference (MD) 0.01 LogMAR (logarithm of the minimal angle of resolution), 95% confidence interval (CI) -0.02 to 0.03; 64 participants)."
A retrospective interventional case series study designed in this way does not change it in this light. This observation is related to CRT, a more interesting observation is the SFCT study after injection of a-VEGF drugs, and this is a potential advantage of this study. The observation of hypertension as a risk factor for CSC refractory is correct, but in my opinion the conclusion as to why this is so is incorrect. There are publications indicating that various drugs, including those for hypertension, block the metabolism of the CYP 3A4 pathway for cortisol, which promotes CSC. Hypertension is such a common disease that the study probably included patients using these drugs. I would suggest the authors fundamental changes to the study, focusing on the SFCT if it is to be retrospective, if it is to be about CRT the evidence must be stronger preferably from prospective cohort studies. Patients should take into account antihypertensive medications that may interfere with the metabolism of the cortisol CYP 3A4 pathway. It should be remembered that the use of injections with poor evidence of their effectiveness may be associated with serious complications such as endophthalmitis and blindness.
Author Response
Response to Reviewers
Journal of Clinical Medicine
Submission ID jcm-1718591
We greatly appreciate the reviewer’s thoughtful critiques of our manuscript. The manuscript has been carefully rechecked, and appropriate changes have been made in RED color in accordance with the reviewer’s comments. Responses to the reviewer’s comments have also been prepared. Our point-by-point responses to reviewer’s comments are given below.
Reviewer Comments and Suggestions for Authors:
Reviewer 1
Reject: The article has serious flaws, does not make an original contribution. Although the topic raised by the authors is interesting and the examination of the thickness of the choroid after the administration of VEGF drugs is important, the article has significant drawbacks. All the meta-analyses show that there is not enough strong evidence that injecting a-VEGF drugs does not have tremendously strong evidence of their effectiveness. Meta-analyzes showed only the efficacy of treatment with mineralocorticoid antagonists and half the dose of PDT.
Salehi M, Wenick AS, Law HA, Evans JR, Gehlbach P. Interventions for central serous chorioretinopathy: a network meta‐analysis. Cochrane Database of Systematic Reviews 2015, Issue 12. Art. No.: CD011841. DOI: 10.1002/14651858.CD011841.pub2. Accessed 15 May 2022."Low quality evidence from two trials suggested little difference in the effect of anti-VEGF (ranibizumab or bevacizumab) or observation on change in visual acuity at six months in acute CSC (mean difference (MD) 0.01 LogMAR (logarithm of the minimal angle of resolution), 95% confidence interval (CI) -0.02 to 0.03; 64 participants)."
Response> Regarding the effect of anti-VEGF in CSC, the article that the reviewer mentioned was a network meta-analysis that included only two trials conducted before 2015, performed to investigate the effect of anti-VEGF in acute CSC. Since the acute form of CSC usually has a benign course with spontaneous resorption of subretinal fluid (SRF), the effect of anti-VEGF may not be evident due to lack of statistical power, when compared with observations. This is why most other studies investigating the effect of anti-VEGF have focused on chronic CSC, in which visual prognosis is usually worse due to longstanding existence of SRF and resulting photoreceptor damage.
Anti-VEGF agents are suggested to be effective for visual and anatomical improvement in chronic CSC by multiple studies [Schaal KB et al. Eur J Ophthalmol 2009;19(4):613-617 / Inoue M et al. Ophthalmologica 2011;225(1):37-20 / Chung YR et al. Retina 2019;39(1):134-142].
Regarding the use of PDT for chronic CSC, we agree that half-dose PDT can be an effective treatment modality for some patients with the disease. However, permanent RPE damage is a significant complication of even half-dose or safety-enhanced PDT in eyes with chronic CSC, which frequently have preexisting RPE and photoreceptor atrophy. [Chan WM et al. Retina 2008;28(1):85-94 / Inoue R et al. Am J Ophthalmol 2010;149(3):441-446]. With respect to the use of mineralocorticoid antagonists, a randomized, double-blind, placebo-controlled trial recently proved that they are not superior to placebo in improving visual acuity in eyes with chronic CSC and strongly recommended that ophthalmologists should discontinue its prescription for this disease. [Lotery A et al. Lancet 2020’395(10220):294-303].
We added this at lines 205–211, page 7.
A retrospective interventional case series study designed in this way does not change it in this light. This observation is related to CRT, a more interesting observation is the SFCT study after injection of a-VEGF drugs, and this is a potential advantage of this study. The observation of hypertension as a risk factor for CSC refractory is correct, but in my opinion the conclusion as to why this is so is incorrect. There are publications indicating that various drugs, including those for hypertension, block the metabolism of the CYP 3A4 pathway for cortisol, which promotes CSC. Hypertension is such a common disease that the study probably included patients using these drugs. I would suggest the authors fundamental changes to the study, focusing on the SFCT if it is to be retrospective, if it is to be about CRT the evidence must be stronger preferably from prospective cohort studies. Patients should take into account antihypertensive medications that may interfere with the metabolism of the cortisol CYP 3A4 pathway. It should be remembered that the use of injections with poor evidence of their effectiveness may be associated with serious complications such as endophthalmitis and blindness.
Response> Thank you for your comment. Due to the retrospective nature of this study, we were unable to investigate the use of antihypertensive medications in patients with hypertension in our population sample. This can be another limitation of this study.
It is known that systemic medications, such as corticosteroids, can induce CSC. Cortisol is metabolized to β-hydroxycortisol by human cytochrome p450-3A4 (CYP3A4), an important enzyme involved in the metabolism of a variety of exogenous and endogenous compounds [Chen Z et al. Steroids 2004;69(1):69-70]. Among antihypertensive medications, calcium channel blockers (CCBs) including verapamil, diltiazem, and amlodipine serve as substrates for and are inhibitors of CYP3A4 [Flockhart DA and Tanus-Santos JE. Arch Intern Med 2002;162(4):105-412 / Zisaki A et al. Curr Phar Des 2015;21(6):806-822]. Taking all of the above together, we agree with the reviewer’s critique that patients using antihypertensive medications affecting systemic cortisol levels might have induced a bias in our study.
However, current trends in the prescription of antihypertensive medications have made ACEIs/ARBs the most commonly prescribed medications, followed by diuretics, β-blockers and, lastly, CCBs [Derington CG et al. Hypertension 2020;75(4):973-981]. CCBs accounted for approximately 1/4th of prescribed antihypertensive medications. Considering this trend, we believe that the use of systemic antihypertensive medications might not have had a large effect on our investigation of anti-VEGF treatment for chronic CSC. However, further prospective studies would be needed to clarify the effects of hypertension and antihypertensive medications on CSC. We added this point at lines 249–257, page 7.
Concerning complications related to intravitreal injection, no serious complications such as endophthalmitis, retinal detachment, or blindness occurred in this study. The risk of endophthalmitis by anti-VEGF injection has been proven to be very low in all related large-scale studies [Bavinger JC et al. Retina 2019;39(10):2004-2011 / Reibaldi M et al. Retina 2018;38(1):1-11].
We wish to thank you for your valuable comments and questions and for giving us the opportunity to strengthen our manuscript. We hope that the revised manuscript is now acceptable to be published in this esteemed journal.

Reviewer 2 Report
This paper presents a retrospective study on changes in choroidal thickness following intravitreal Bevacizumab injections in chronic CSC.
The manuscrip is clearly understandable and the retrospective study is well conducted and documented.
I have only few suggestions to the Authors:
- 1 abstract line 16 to 18 the adverb significantly is repeated 3 times in e lines, maybe could be replaced by other synonyms
- tex line 39 please better explain the reference number 6 . What does " higher vascuar luminar ratio" mean?
- text line 229-231 please add citations to the sentences.
Author Response
Response to Reviewers
Journal of Clinical Medicine
Submission ID jcm-1718591
We greatly appreciate the reviewer’s thoughtful critiques of our manuscript. The manuscript has been carefully rechecked, and appropriate changes have been made in RED color in accordance with the reviewer’s comments. Responses to the reviewer’s comments have also been prepared. Our point-by-point responses to reviewer’s comments are given below.
Reviewer Comments and Suggestions for Authors:
Reviewer 2
This paper presents a retrospective study on changes in choroidal thickness following intravitreal Bevacizumab injections in chronic CSC.
The manuscrip is clearly understandable and the retrospective study is well conducted and documented.
I have only few suggestions to the Authors:
- 1 abstract line 16 to 18: the adverb significantly is repeated 3 times in e lines, maybe could be replaced by other synonyms
Response> We appreciate the reviewer’s comment. We changed the sentences in abstract as follows: “SFCT was significantly decreased along with the CRT following IVB at the resolved state. SFCT reduction following 1 month of IVB was notably greater in the resolved eyes. The association of refractory eyes with hypertension (p = 0.003) and a thinner baseline SFCT (p = 0.024) was significant. In most of the patients with chronic CSC, VA and CRT improved remarkably following treatment with IVB.
- text line 39 please better explain the reference number 6: What does " higher vascuar luminal ratio" mean?
Response> Thank you for your suggestion. Lee et al. [6] reported that a significant reduction in stromal area was noted in patients with CSC along with increased luminal area (i.e., higher vascular luminal-to-total choroidal ratio). The choroidal vascular area (luminal area) positively correlated with the total choroidal thickness in CSC; thus, the authors suggested that chronic venous congestion and subsequent stromal tissue atrophy might be involved in the pathophysiology of the disease. We added this point at lines 39–44, page 1 for more clarity.
- text line 229-231 please add citations to the sentences.
Response> Thank you for your comment. We added the citation of the review article by Daruich et al. [1], which summarized the OCT findings of choroid in CSC (lines 241–243, page 7).
We wish to thank you for your valuable comments and questions and for giving us the opportunity to strengthen our manuscript. We hope that the revised manuscript is now acceptable to be published in this esteemed journal.
Reviewer 3 Report
The authors investigated the change in the subfoveal choroidal thickness (SFCT) in eyes with chronic CSC after IVB. Overall, it is an interesting and well-written paper.
1. In abstract and Line 80, What does it mean by ‘1 month following initial IVB’? How many IVB were given by then?
2. Based on the methods, I would prefer to make it clear in the results, Line 107 that these eyes with chronic CSC were treatment-naïve and without macular neovascularization, which will have an effect on the choroid as well.
3. For those refractory CSC, the follow-up time was around 16.9m. Why was the PDT not initiated in these eyes? Also, please specify the rationale in the Methods for using IVB only to treat chronic CSC.
4. Did authors investigate the choroidal vascularity index (CVI), as other papers did?
PMID: 26868048. If not, please mention it in the limitation.
5. In the discussion, please speculate the mechanism of SFCT change after IVB in CSCR? Does it have a similar mechanism as that proposed in PCV and AMD? PMID: 34860239; PMID: 26743619
Author Response
Response to Reviewers
Journal of Clinical Medicine
Submission ID jcm-1718591
We greatly appreciate the reviewer’s thoughtful critiques of our manuscript. The manuscript has been carefully rechecked, and appropriate changes have been made in RED color in accordance with the reviewer’s comments. Responses to the reviewer’s comments have also been prepared. Our point-by-point responses to reviewer’s comments are given below.
Reviewer Comments and Suggestions for Authors:
Reviewer 3
The authors investigated the change in the subfoveal choroidal thickness (SFCT) in eyes with chronic CSC after IVB. Overall, it is an interesting and well-written paper.
- In abstract and Line 80, What does it mean by ‘1 month following initial IVB’? How many IVB were given by then?
Response> We apologize if the meaning of the phrase was not clear in the original manuscript. The phrase ‘”1 month following initial IVB” meant that the first efficacy evaluation of visual acuity, central retinal thickness, and SFCT was performed 1 month after the first IVB treatment in treatment-naïve patients.
- Based on the methods, I would prefer to make it clear in the results, Line 107 that these eyes with chronic CSC were treatment-naïve and without macular neovascularization, which will have an effect on the choroid as well.
Response> Thank you for your comment. We added the suggested explanation to lines 114–115, page 3.
- For those refractory CSC, the follow-up time was around 16.9m. Why was the PDT not initiated in these eyes? Also, please specify the rationale in the Methods for using IVB only to treat chronic CSC.
Response> Unfortunately, PDT was not available in our country during most of the period covered by our study, when the included patients received IVB treatment, for the following reasons: 1) Maintenance support for the PDT machine from the manufacturer was not available, and 2) verteporfin dye was not imported.
Moreover, PDT has potential risk of complications, such as permanent RPE atrophy or choriocapillaris hypoperfusion [Chan WM et al. Retina 2008;28(1):85-93 / Inoue R et al. Am J Ophthalmol 2010;149(3):441-446]. As patients with chronic CSC are already prone to have RPE atrophy, PDT was our last option for these patients.
We added this rationale to the Discussion section, rather than in the Methods section (lines 269–274, page 8).
- Did authors investigate the choroidal vascularity index (CVI), as other papers did?
PMID: 26868048. If not, please mention it in the limitation.
Response> We could not investigate the choroidal vascularity index in this study. As the suggested by the reviewer, we added this as limitation at lines 275–276, page 8 citing reference PMID 26868048.
- In the discussion, please speculate the mechanism of SFCT change after IVB in CSCR? Does it have a similar mechanism as that proposed in PCV and AMD? PMID: 34860239; PMID: 26743619
Response> Thank you for your comment. The articles that the reviewer mentioned suggest the role of anti-VEGF therapy on the choroidal vascular flow and volume, leading to reduced choroidal thickness. We added this hypothesis of anti-VEGF therapy in PCV at lines 229–230, page 7.
Similarly, anti-VEGF therapy might reduce choroidal hyperpermeability with subsequent reduction in choroidal thickness, as we had already mentioned at lines 212–215, page 7.
We wish to thank you for your valuable comments and questions and for giving us the opportunity to strengthen our manuscript. We hope that the revised manuscript is now acceptable to be published in this esteemed journal.

Round 2
Reviewer 1 Report
Although some studies have investigated the use of anti-VEGF for CSC, no large, prospective randomised controlled clinical trials have been performed and I am still not convinced to use anti-VEGF for CSC. The introduced correction enables publication. I still believe that authors should focus on choroidal thickness rather than the efficacy of ant-VEGF therapy for CSC
Author Response
We greatly appreciate the reviewer’s thoughtful critiques of our manuscript. The manuscript has been carefully rechecked, and appropriate changes have been made in RED color in accordance with the reviewer’s comments. Responses to the reviewer’s comments have also been prepared. Our point-by-point responses to reviewer’s comments are given below.
Response> We appreciate your insightful comment. We agree that the lack of randomized controlled studies is critical to determine the effect of anti-VEGF agents in CSC. Along with the reviewer’s comment, we added the following sentences as limitation in Discussion section at page 8, lines 269-274:
“It should be noted that there are still no prospective randomized controlled clinical trials with large sample size that investigated the effect of anti-VEGF agents in CSC. Accordingly, caution should be taken in the interpretation of the results although we found a positive effect of IVB in chronic CSC. Meanwhile, a more interesting point of this retrospective study is that the change of choroidal thickness following IVB was associated with the therapeutic response in chronic CSC, and this is a potential advantage of this study.”
We wish to thank you for your valuable comments and for giving us the opportunity to strengthen our manuscript. We hope that the revised manuscript is now acceptable to be published in this esteemed journal.
